# The Raw Milk Microbiota from Semi-Subsistence Farms Characteristics by NGS Analysis Method

**DOI:** 10.3390/molecules26165029

**Published:** 2021-08-19

**Authors:** Bartosz Hornik, Jakub Czarny, Justyna Staninska-Pięta, Łukasz Wolko, Paweł Cyplik, Agnieszka Piotrowska-Cyplik

**Affiliations:** 1Institute of Forensic Genetics, Al. Mickiewicza 3/4, 85-071 Bydgoszcz, Poland; b.hornik@igs.org.pl (B.H.); pubjc@igs.org.pl (J.C.); 2Department of Food Technology of Plant Origin, Poznan University of Life Sciences, Wojska Polskiego 31, 60-624 Poznań, Poland; justyna.staninska@up.poznan.pl; 3Department of Biochemistry and Biotechnology, Poznan University of Life Sciences, Dojazd 11, 60-632 Poznań, Poland; lukasz.wolko@up.poznan.pl; 4Department Biotechnology and Food Microbiology, Poznan University of Life Sciences, Wojska Polskiego 48, 60-627 Poznań, Poland; pawel.cyplik@up.poznan.pl

**Keywords:** antibiotic resistance, raw milk microbiome, farm, next-generation sequencing

## Abstract

The aim of this study was to analyze the microbiome of raw milk obtained from three semi-subsistence farms (A, B, and C) located in the Kuyavian-Pomeranian Voivodeship in Poland. The composition of drinking milk was assessed on the basis of 16S rRNA gene sequencing using the Ion Torrent platform. Based on the conducted research, significant changes in the composition of the milk microbiome were found depending on its place of origin. Bacteria belonging to the *Bacillus* (17.0%), *Corynebacterium* (12.0%) and *Escherichia-Shigella* (11.0%) genera were dominant in the milk collected from farm A. In the case of the milk from farm B, the dominant bacteria belonged to the *Acinetobacter* genus (21.0%), whereas in the sample from farm C, *Escherichia-Shigella* (24.8%) and *Bacillus* (10.3%) dominated the microbiome. An analysis was performed using the PICRUSt tool (Phylogenetic Investigation of Communities by Reconstruction of Unobserved States) in order to generate a profile of genes responsible for bacterial metabolism. The conducted analysis confirmed the diversity of the profile of genes responsible for bacterial metabolism in all the tested samples. On the other hand, simultaneous analysis of six KEGG Orthologs (KO), which participated in beta-lactam resistance responsible for antibiotic resistance of bacteria, demonstrated that there is no significant relationship between the predicted occurrence of these orthologs and the place of existence of microorganisms. Therefore, it can be supposed that bacterial resistance to beta-lactam antibiotics occurs regardless of the environmental niche, and that the antibiotic resistance maintained in the population is a factor that shapes the functional structure of the microbial consortia.

## 1. Introduction

Milk and its products play an important role in human nutrition in many cultures. It is a source of protein, vitamins and many minerals, which makes it an excellent environment for the growth and development of bacteria [1,2]. Raw cow milk is one of the most diverse raw materials in terms of microbiology, which directly affects the quality and price of manufactured products and the company’s financial results [3]. The colonization of milk by microorganisms is a significant threat that has a negative impact not only on the quality and durability of products, but also on human health [4]. In order to achieve a positive effect of the health properties of milk on the human body, appropriate hygienic conditions should be maintained not only during collection, but also during its transport to dairy plants, processing and preservation.

Milk, in addition to its endogenous microbiota, enables the development of microorganisms which may originate from the surface of animals, as well as from the environment in which the livestock lives [4]. The contamination of milk with bacteria mainly results from poorly cleaned and poorly disinfected milking equipment, lack of hygiene during milking or handling of the milk, and bacteria present in the barn. Moreover, the presence of environmental pathogens such as *Escherichia coli*, *Klebsiella* spp., *Streptococcus dysgalactiae* and *Streptococcus uberis* may contribute to the development of mastitis in dairy cattle, which is an “occupational disease” of high-yielding dairy cows [5,6]. The disease is caused by bacteria and other microorganisms that enter the teats. The appearance of this disease in high-yielding dairy cows is the cause of huge economic losses [7]. There are two forms of mastitis: clinical and subclinical, which is a latent form. It is estimated that up to 50% of cows may suffer from subclinical mastitis. Unfortunately, in the subclinical form, there are also economic losses caused by the reduced milk yield of cows; hence, the correct diagnosis and implementation of an effective treatment process are very important [8]. 

The treatment of mastitis in cows, despite the implementation of prophylactic programs and new methods of therapy, is still mainly based on the administration of antibiotics. It is estimated that animal production in most countries sometimes accounts for as much as 80% of the total consumption of antibiotics [9]. Unfortunately, their presence in food has negative health and economic consequences for humans [10]. One of the greatest threats to public and global health associated with the use of antibiotics is the increase in antibiotic resistance of bacteria [11]. It is assumed that the main cause of this phenomenon is their excessive use, and the scale of abuse in this area makes them the main cause of the growth and spread of antibiotic-resistant bacteria and resistance genes in the environment [12]. 

Antibiotics used to treat and prevent bacterial infections in animals may also contribute to the formation of drug-resistant bacterial strains in the human body [13]. It can be said with certainty that the process of managing antibiotic resistance has progressed considerably, meaning that it is leading the world into the post-antibiotic era. Unfortunately, the alarming forecasts from scientists have become completely realistic, and harmless bacterial infections that have been successfully treated in the past appear to be deadly today [14]. 

Among the antibiotics, more than 60% of the intramammary preparations used contain β-lactams (penicillins, cephalosporins), which is confirmed by the results of the presence of antibiotics in milk. One of the main reasons for the insensitivity to antibiotics is the ability of bacteria to produce β-lactamase, an enzyme that neutralizes the action of, for example, penicillins and cephalosporins [12]. On the one hand, the subthreshold concentrations of antibiotics strongly influence the selection of resistant strains, and they contribute to the formation of morphologically changed bacteria [15,16]. In addition, the environment in which antibiotics are present favors the transfer of mobile genetic elements by horizontal gene transfer, which leads to the dissemination of resistance genes even between phylogenetically distant bacteria [17]. 

Due to the above-mentioned reasons, the assessment of the composition and direction of development of the microbiota present in raw milk has a significant impact not only on the composition and quality of milk, but above all on the quality of dairy products [18]. The microorganisms present in these products may affect human health. They constitute a reservoir of genes that determine the antibiotic resistance of bacteria, which can be permanently transferred to the microbiome of the human gastrointestinal tract [19]. On this basis, sequence analysis of the hypervariable regions of the 16S rRNA gene using the Ion Torrent platform (Life Technologies, Carlsbad, CA, USA) was used to assess the microbiome of milk from semi-subsistence farms. Then, a functional analysis of the milk microbiome was carried out based on the PICRUSt tool in order to determine the potential of the microbiome as a carrier of genes that determine the resistance of bacteria to β-lactam antibiotics.

## 2. Results and Discussion

### 2.1. Taxonomic Analysis of the Milk Microbiome 

Milk is an excellent environment for the development of microorganisms responsible for the specific properties of many dairy products as well as microbes which are undesirable for technological and health reasons (review by Quigley et al. [3]). The use of next-generation sequencing allowed to detect DNA from bacteria, the presence of which could not be confirmed to date by cultivation-based methods [17,20,21]. The taxonomic identification carried out on the basis of the sequence analysis of the hypervariable regions of the 16S rRNA gene based on the SILVA v119 database enabled the detection of microorganisms that are components of milk collected from cows from semi-subsistence farms.

In all of the studied samples, Firmicutes was the dominant phylum and its ratio in the milk microbiome ranged from 42% to 53%. The other types present in milk included *Proteobacteria* (40–26%), *Actionobacteria* (14–5%) and *Bacteroidetes* (10–3%). All samples displayed the presence of bacteria belonging to 42 classes (Figure 1).

Five classes of bacteria were found to be dominant in the milk samples collected from all of the researched farms: Gammaproteobacteria (21.93–42.77%), Bacilli (17.88–36.26%), Clostridia (7.04–14.88%), Actinobacteria (5.14–13.82%) and Betaproteobacteria (1.56–7.28%).

### 2.2. Lactic Fermentation Bacteria

The newly carried out generation sequencing allowed for species and functional identification of lactic acid bacteria (LAB). In all samples, different amounts of lactic acid bacteria (LAB) were found, which belonged to the following types: *Carnobacterium*, *Enterococcus*, *Lacticigenium*, *Lactobacillus*, *Lactococcus*, *Leuconostoc*, *Streptococcus* and *Trichococcus*. The highest ratio of LAB bacteria in the milk microbiome was found in the sample collected from farm C and it was equal to 9.62%. In other farms, it ranged from 5.2% to 6.1%.

LABs are the most important group of microorganisms found in milk, because their role is to convert carbohydrates and proteins into numerous secondary metabolites [9]. The compounds produced by these bacteria stimulate the growth of other groups of microorganisms, and thus directly and indirectly determine the final shape of the finished product [22]. 

### 2.3. Spoiled Milk Bacteria

Some scientists have suggested that the bacteria found in milk originate not only from external sources, but also enter the milk as a result of migration from other internal organs through the so-called internal colonization. The presence of bacteria belonging to the Ruminococcus and Bifidobacterium genera, as well as to the *Peptostreptococcaceae* family, in the analyzed milk samples was first described by Young et al. [23], who showed in their research that these bacteria can escape from the intestinal lumen and travel through the mesentery lymph nodes to the mammary gland. 

The higher sensitivity of the used molecular methods compared to the cultivation-based methods revealed the presence of numerous microorganisms as responsible for the spoilage of milk. A large group identified in all of the analyzed milk samples were Gram-negative bacteria belonging to the *Pseudomonas*, *Acinetobacter* and *Aeromonas* genera [24]. Such bacteria are characterized by the ability to produce lipases, which are responsible for unfavorable changes in the taste and smell of milk. The milk also contains coliform bacteria, which are characterized by the ability to ferment lactose, and Gram-positive spore bacteria (*Bacillus*) responsible for milk coagulation. The presence of these groups of bacteria is confirmed by numerous previous studies of the milk microbiota [20,21]. 

Studies of the milk microbiota based on next-generation sequencing also identified anaerobic bacteria such as *Bacteroides*, *Faecalibacterium*, *Prevotella* and *Catenibacterium,* the presence of which may be related to the presence of fecal contaminants.

### 2.4. Pathogenic Bacteria Causing Mastitis

A large group of microorganisms present in milk samples included pathogenic and potentially pathogenic bacteria responsible for the occurrence of mastitis in dairy cattle. The identified pathogenic bacteria belonged to the Proteobacteria and Firmicutes phyla. There are several pathogens that most often cause inflammation of the mammary glands in cows: *Staphylococcus aureus*, *Escherichia coli*, *Streptococcus agalactiae*, *Streptococcus dysgalactiae*, *Streptococcus uberis*, coagulase-negative *Staphylococci*, *Enterococcus* spp. (mainly *E. faecium* and *E. faecalis*), and their prevalence differs in every country and even in every herd that are tested. Proteobacteria are a diverse taxonomic unit that covers a wide group of pathogens. They are Gram-negative bacteria considered to be environmental mastitis pathogens, in contrast to Gram-positive bacteria belonging to Firmicutes, which are considered contagious mastitis pathogens [25,26,27]. 

However, in terms of the generic composition of bacteria, significant differences were found between the analyzed samples collected from the researched farms (Figure 2). Bacteria belonging to the *Bacillus* (17%), *Corynebacterium* (12%) and *Escherichia-Shigella* (11%) genera dominated in the milk collected from farm A. In the milk from farm B, bacteria belonging to the *Acinetobacter* genus were dominant (21%), whereas in the sample from farm C, *Escherichia-Shigella* (24.8%) and *Bacillus* (10.3%) were the most abundant members. The ratio of other types of bacteria did not exceed 10% in all of the samples collected from farms A, B and C. 

The dominant bacterium was *Escherichia coli*. Among the remaining 20 dominant bacteria, as many as 12 were uncultured bacteria, which cannot be identified and diagnosed using classical microbiology methods. The other dominant bacteria included: *Achromobacter xylosoxidans*, *Streptococcus pyogenes*, *Streptococcus pseudoporcinus*, *Bacillus subtilis i Chryseobacterium hagamense*, *Corynebacterium bovis*, *Aerococcus suis*, and *Corynebacterium sphenisci*.

### 2.5. Analysis of Biodiversity

The values of the analyzed alpha-biodiversity coefficients in the metapopulation of the microorganisms present in milk are presented in Table 1. Estimates of intrasample diversity were carried out at a rarefaction depth of 100,000 reads per sample. There was a significant difference in the number of identified OTUs in all tested samples. Moreover, almost all of the alpha-biodiversity indices differ from each other (except the Shannon entropy, which showed no significant difference between milk from farm B and C). In all of the determined indicators, the microbiome of milk obtained from farm A displayed the highest biodiversity.

The results of the beta-biodiversity analysis of variants collected from three farms are presented in Figure 3.

The same relationships were observed between the majority of the studied variables for both of the methods of scaling of three principal components in space. The location of points from the individual milk samples in separate spaces on the chart is consistent with their origin, which indicates significant differences in the bacterial composition related to the place of their collection. This is confirmed by the predictions of the functional metabolic metapopulation profiles carried out overall (Figure 4).

### 2.6. Analysis of Functional Potential and Antibiotic Resistance

Based on the PCA analysis and the structure of the dendrogram, there are significant differences in the overall predicted functional potential between the populations of microorganisms collected from the different farms. There is a clear influence of the ecological niche on the shaping of the structure of the predicted (putative) bacterial metagenome [28].

The treatment of mastitis in cows, despite the implementation of prophylactic programs and new genetic methods of diagnosis and treatment (such as sequencing for identification of pathogens and screening for antibiotic resistance genes), is still mainly based only on the administration of antibiotics, despite many negative aspects of their use and inconsistent satisfactory effectiveness [29,30]. One of the main reasons for the lack of sensitivity to antibiotics is the ability for bacteria to produce enzymes (e.g., β-lactamase) that neutralize the activity of selected antibiotics [31]. The genes responsible for the production of β-lactamases are found either on the bacterial chromosome or on the plasmid [32].

The production of various β-lactamases and acylases by Gram-positive and Gram-negative bacteria, which remove the side-substituent of the antibiotic, is the key cause of resistance to β-lactam antibiotics [33].

The variety of bacterial taxa, in which the presence of genes participating in the beta-lactam resistance pathway was predicted in this study, indicates the importance of the phenomenon of horizontal transfer of this trait. It also seems that the features of resistance to beta-lactam antibiotics occur regardless of the environmental niche and are an important factor which shapes the functional structure of microbial consortia. 

The analysis, limited to six KEGG Orthologs (KO) that participated in beta-lactam resistance, showed that there is no significant relationship between the predicted occurrence of these orthologs and the place of microbial existence (Figure 5). A detailed, predicted representation of these genes in individual phyla and bacterial classes is presented in Figure 6. The functional features related to resistance to beta-lactam antibiotics are not limited to a few taxonomic groups. Nevertheless, the following phyla in which the probability of these orthologs is particularly high can be distinguished: Proteobacteria, Firmicutes, Bacterioidetes, Actinobacteria, Acidobacteria and Cyanobacteria. Moreover, the PCA analysis showed that beta-lactam resistance functional metabolic profiles did not differ significantly between the populations of microorganisms isolated from the different farms (Figure 6).

## 3. Materials and Methods

### 3.1. Characteristics of Farms and Collection of Samples

The microflora of milk from three farms located in Poland in the Kuyavian-Pomeranian voivodship (farms A, B and C) was subjected to metagenomic analysis. The farms were semi-subsistence farms with 5 to 8 dairy cows. A 500 mL sample of the bulk milk was collected from each farm every week. The samples collected during each month were mixed with each other, and thus, from February to November 2020, 10 averaged samples from each farm were prepared. In total, 30 samples were analyzed. The samples were collected in sterile containers and transported to the microbiological laboratory at 4 °C, and then stored at −20 °C. 

### 3.2. Isolation of DNA

DNA isolation from milk samples was performed using the Genomic Mini AX Bacteria Spin kit (060-100S, A&A Biotechnology, Gdańsk, Poland) according to the protocol provided by the manufacturer. Finally, the purified DNA was eluted. The isolates were stored at −80 °C after they had been neutralized, in order to minimize matrix degradation.

The efficiency of isolation was checked each time based on the fluorimetric method with the use of the Qbit 3.0 device and the Qubit™ dsDNA HS Assay Kit (Q32851, ThermoFisher Scientific, Waltham, MA, USA). For each sample, three DNA extractions were performed and finally combined after a positive quantification.

### 3.3. PCR Amplification and NGS Sequencing

The PCR reaction was prepared using the Ion 16S™ Metagenomics Kit (A26216, Life Technologies). This kit allows for the amplification of the V2–V9 regions of the bacterial 16S rRNA gene. The reaction was prepared according to the manufacturer’s instructions. The reaction consists of 15 µL of 2× Environmental Master Mix, 3 µL of the appropriate primer and 12 µL of the DNA sample previously isolated from the milk sample. The reaction was performed in a Veriti thermal cycler (Life Technologies) using the following temperature program: initial denaturation for 10 min at 95 °C; 25 cycles of denaturation for 30 s at 95 °C; annealing for 30 s at 58 °C; extension for 20 s at 72 °C; and a final extension for 7 min at 72 °C.

The reaction products were purified using the Agencourt AMPure XP Reagent (A63880, Beckman Coulter, Pasadena, CA, USA), according to the manufacturer’s instructions. The method was based on binding DNA to magnetic beads followed by washing away the contaminants with ethanol. The DNA was rinsed from the beads using nuclease-free water or low-TE buffer. A library was prepared according to the manufacturer’s instructions using the Ion Plus Fragment Library Kit (4471252, Life Technologies). The prepared library was purified using Agencourt AMPure XP Reagent (A63880, Beckman Coulter, Pasadena, CA, USA), according to the manufacturer’s instructions. The concentration of the library was assessed using the Ion Universal Library Quantitation Kit and a real time PCR instrument—Quant Studio 5 (A26217, Life Technologies). The library was then diluted to a concentration of 10 pM. The diluted library was coated onto beads (used for sequencing) in emulsion PCR using the Ion PGM™ Hi-Q™ View OT2 Kit reagent kit and an Ion One Touch 2 Instrument (A29900, Life Technologies). The library-coated beads were purified using an Ion One Touch ES Instrument (Life Technologies).

The library-coated beads were sequenced using an Ion PGM System (Life Technologies) using the Ion PGM™ Hi-Q™ View Sequencing Kit (A29900) on an Ion 316™ Chip Kit v2 BC.

The 16S rRNA sequencing datasets generated and analyzed during the current study have been deposited at the National Center for Biotechnology Information (SRA repository), as BioProject under ID PRJNA699887.

### 3.4. Bioinformatic Analysis

The sequence reads from the Ion Torrent (Thermo Fisher Scientific) in BAM format were imported into the CLC Genomics Workbench 20.0 software (Qiagen, Hilden, Germany) and processed with CLC Microbial Genomics Module 20.1.1 (Qiagen, Hilden, Germany). The total number of reads and results of downstream processing for all samples were presented in the Appendix A (Appendix A). Chimeric and low-quality reads (quality limit = 0.05, ambiguous limit = ‘N’) were filtered and removed. Then, the sequence reads were clustered against the SILVA v119 [34] database at 97% similarity of operational taxonomic units (OTU). Finally, the merged abundance table was generated, and selected alpha (number of OTUs, Chao-1 bias-corrected, Shannon entropy and Phylogenetic diversity) and beta (Bray–Curtis principal coordinate analysis) diversity parameters were determined. 

### 3.5. Prediction the Functional Profile from Targeted Metagenomic Data

In order to generate a profile of the putative functional properties of the analyzed microbial consortia based on the targeted 16S rRNA OTU data, an analysis was carried out using the PICRUSt (ver.1) (Phylogenetic Investigation of Communities by Reconstruction of Unobserved States) tool [35]. The OTUs derived from clustering against GreenGenes 13.5 (97% similarity), which were normalized for 16S rRNA copy numbers, were used as input data. A total of 6909 KEGG Orthologs (KO) were identified (translate as abundance), 6 of which were related to beta lactam-resistance (Table 2). PCA visualization and dendrogram analysis (Jensen–Shannon divergence) of the obtained data were performed using the MicrobiomeAnalyst software (McGill University, Montreal, Canada) [36].

### 3.6. Statistical Analysis

Statistical analysis was performed using the Statistica software (StatSoft, ver.13.3). Error margin ranges represent standard errors of the mean and were calculated by dividing the standard deviation by the square root of the sample size. The Wilcoxon rank sum test (*p* < 0.05) was used to conduct multiple nonparametric pairwise comparisons after Kruskal–Wallis rank sum tests (*p* < 0.05), to compare the differences among bacterial metapopulations. 

## 4. Conclusions

Microorganisms that developed mechanisms which allow them to survive in the presence of an antimicrobial agent, using the possibilities offered by horizontal gene transfer, can colonize other organisms or transfer their resistance genes to other bacteria. The development of new bioinformatics tools allows for a more accurate description of the diversity of the milk microbiota and the association of taxonomic, physiological and functional characteristics, which allows the dairy industry to improve the quality of milk and dairy products. In the presented research, it was found that the species composition of the milk microbiome from three different farms differed significantly. This differentiation was confirmed by the predicted functional analysis of the genetic potential of the bacteria responsible for their metabolism. However, such differences were not shown by the prediction analysis of genes responsible for antibiotic resistance. Therefore, it can be concluded that a reservoir of genes which determine the resistance to antibiotics is supposed to exist and maintained in the population of microorganisms. There is a high probability that bacteria present in milk constitute an excellent reservoir of resistance genes for potentially harmful bacteria, which poses a serious threat to humans and a medical challenge. Furthermore, the increased possibility of the presence of resistant bacteria in the gut flora results in a higher probability of transferring the resistance genes to (potentially) pathogenic bacteria, as well as their distribution in the environment, and their distribution from animals to food of animal origin.

## Figures and Tables

**Figure 1 molecules-26-05029-f001:**
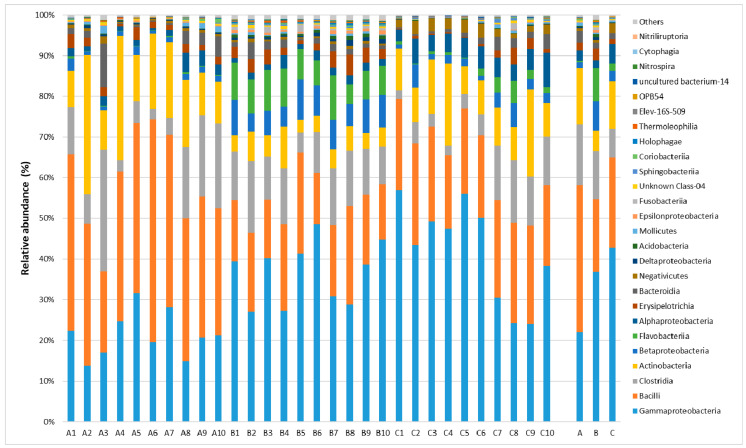
The relative abundances of bacterial classes in milk collected from semi-subsistence farms (A1—A10, B1—B10, C1—C10 represent the individual samples from respective farms; A, B, C represent the mean value for all samples for each farm).

**Figure 2 molecules-26-05029-f002:**
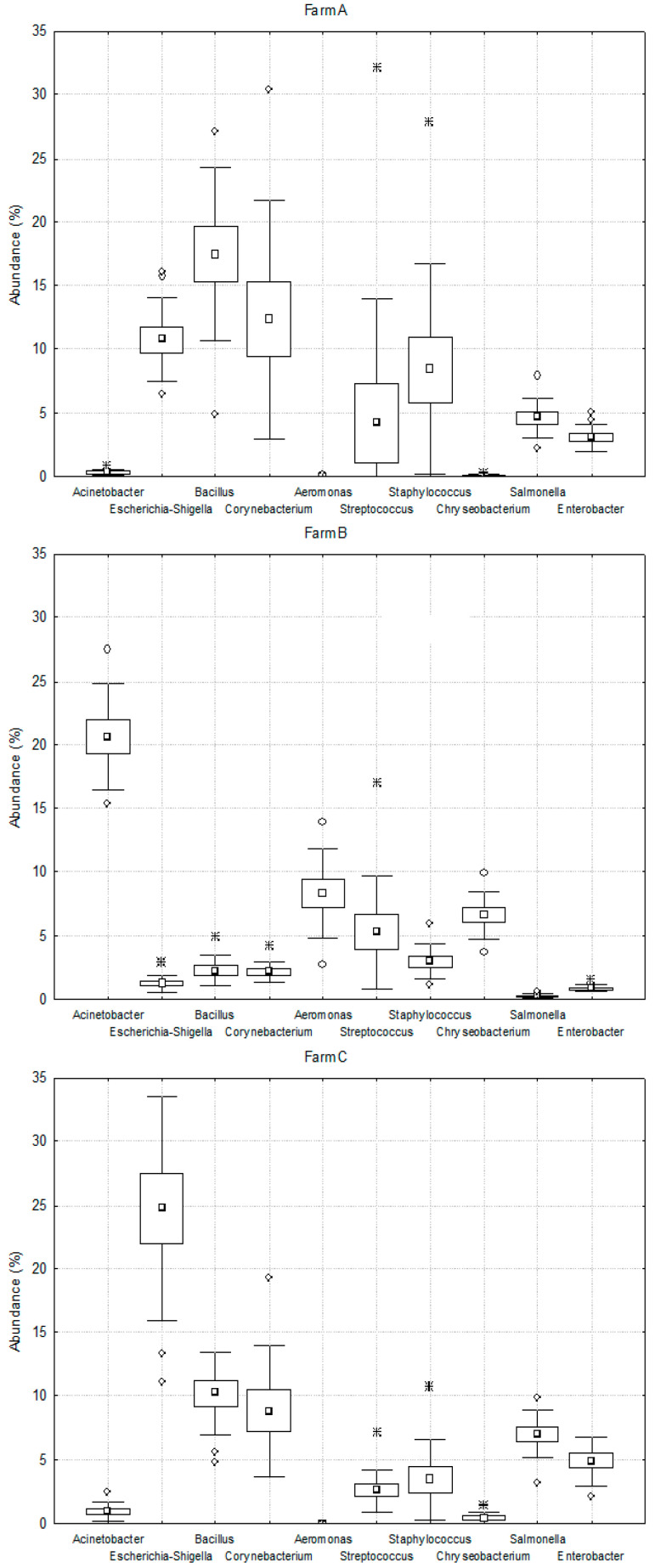
The relative abundance of bacterial genera in milk collected from semi-subsistence farms (A, B, C—milk from respective farms). 

—mean, 
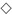
—outlers point, ∗—extreme point.

**Figure 3 molecules-26-05029-f003:**
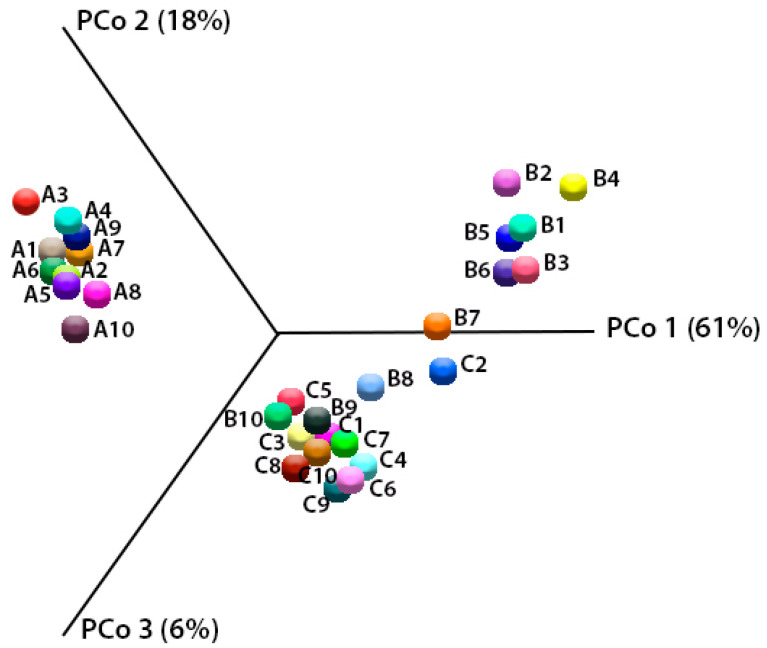
Principal coordinate analysis (PCoA) based on the Bray–Curtis dissimilarity metrics showing the distance in the bacterial communities between analyzed samples (milk from farm A—A1, A2…; milk from farm B—B1, B2…; milk from farm C—C1, C2….).

**Figure 4 molecules-26-05029-f004:**
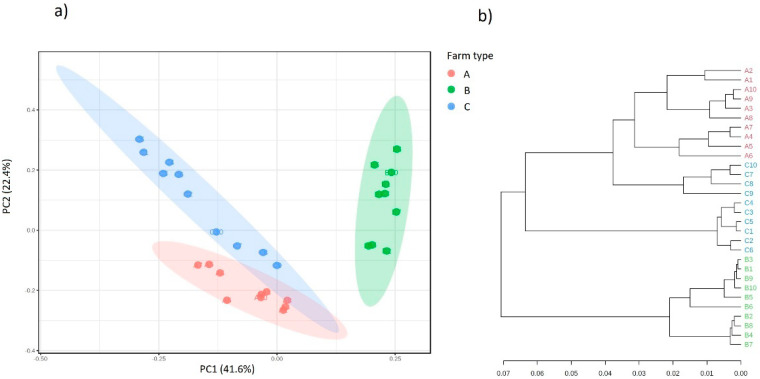
Clustering analysis of the overall, functional metabolic profile of bacterial communities in the analyzed milk samples derived from farms A, B and C (**a**) principal coordinate analysis (PCA) and (**b**) dendrogram analysis showing the distance in the predicted functional metabolic profile between samples.

**Figure 5 molecules-26-05029-f005:**
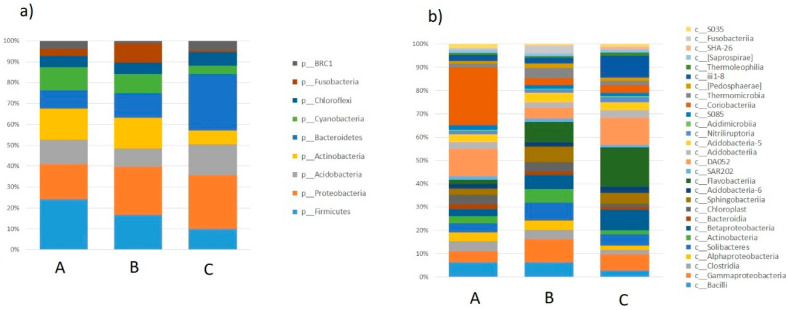
The relative representation of beta-lactam resistance gene orthologs in (**a**) individual classes and (**b**) phyla of bacteria in analyzed samples derived from farms A, B and C.

**Figure 6 molecules-26-05029-f006:**
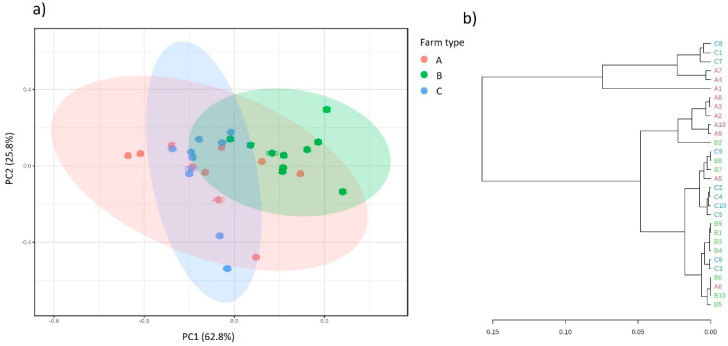
Clustering analysis of beta-lactam resistance functional metabolic profile of bacterial communities in the analyzed milk samples derived from farms A, B and C: (**a**) principal coordinate analysis (PCA) and (**b**) dendrogram analysis showing the distance in the predicted functional metabolic profile between samples.

**Table 1 molecules-26-05029-t001:** Analysis of the alpha-biodiversity of bacteria present in the analyzed milk samples.

	Milk from Farm A	Milk from Farm B	Milk from Farm C
Number of OTUs	5490 ± 78	4486 ± 94	3488 ± 58
Chao-1 bias-corrected	5503 ± 14	4597 ± 18	3508 ± 14
Shannon entropy	8.59 ± 0.45	7.02 ± 0.36	6.33 ± 0.39
Phylogenetic diversity	10.23 ± 0.61	8.98 ± 0.49	7.12 ± 0.44

**Table 2 molecules-26-05029-t002:** List of KEGG gene orthologs (KO) analyzed with the PICRUSt tool related to beta-lactam resistance.

KO Identifier	Gene Name
K01467	beta-lactamase class C (ampC)
K02171	BlaI family transcriptional regulator, penicillinase repressor (blaI)
K02172	bla regulator protein blaR1 (blaR1)
K02545	penicillin-binding protein 2 prime (mecA)
K02546	BlaI family transcriptional regulator, methicillin resistance regulatory protein (mecI)
K02547	methicillin resistance protein (mecR1)

## Data Availability

The 16S rRNA sequencing datasets generated and analyzed during the current study have been deposited at the National Center for Biotechnology Information (SRA reposito-ry), as BioProject under ID PRJNA699887.

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
