# Peer review of "The Raw Milk Microbiota from Semi-Subsistence Farms Characteristics by NGS Analysis Method"

_molecules, 2021, doi:10.3390/molecules26165029_

Round 1
Reviewer 1 Report
Please see below comments regarding your manuscript focused on taxonomic composition of milk samples collected from three different farms. Overall, the manuscript needs improvement and revision.
- Conclusions in abstract and in the conclusion section of the manuscript are not justified by the data because all resistome data are provided by predicted analysis of bacterial function.
- Introduction: mastitis section of the manuscript is not relevant to the manuscript, since the samples were collected from healthy animals and the disease state was not a factor in the analysis.
- Since no resistome analysis was performed, all antibiotic resistance genes data are only speculative and based only on predicted function of microbiota based on 16S.
- There were two different databases used for the analysis, SILVA for 16S and Grengenes for PICRUST. Why? One database should be used for both analysis to relate these two data sets and results.
- The results section lacks information about sequencing results: number of raw reads, average reads per sample, how many reads were removed, rarefaction curve, sequencing depth used for analysis, etc.
- Lack of statistical analysis description for alpha and beta diversity as well as for taxonomic profiling.
- Was the relative abundance of specific bacteria significantly different between farms? The authors may consider running LEfSe analysis to determine differentially abundant bacteria between farms.
- Please provide statistical analysis for section 2.2 and 2.3. It looks like these parts are only discussion with no real results.
- All Figure legends need to be revised to provide more information. If two or more panels are part of one figure, they need to be labeled a, b, etc., and their description should be provided.
- Figure 2: the comparison should be between farms.
- There are two tables labeled “Table 1” (page 7 and 11).
- Table 1, page 7: please provide if alpha diversity indices were significant between farms.
- 3: legend does not correspond to the figure.
- Line 282: replace washed away with eluted
- Methods: if the manufacturer protocol was fallowed to the dot, there is no need for detailed description of the method.
- Which PICRUSt 1 or 2 was used for analysis?
- Overall, there is too much focus on resistome without really doing the resistome analysis and relying only on predicted function.
Reviewer 2 Report
- graphs in the figure 2 (y axis) has to be starded from 0. Please correct them.
- in general figures have to be put in better quality; now is very small font and there are unreadable.
- Please explain in the footnotes under each figure, table all abbreviations. They have to be self-explanatory.
- conclussion should be shortened.
- references need be rewritten according to Journal`s recommendation.
- some typo mistakes should be eliminated.
- please give the catalog number for all used reagents.
- Authors should discuss not only about the microorganisms in the context of beta-lactamase, but also other antibiotics. They should discuss about natural substances - cordyceps, betulin, and a antibiotics used in oncology - salinomycin.
- the paper has to be improved.
Reviewer 3 Report
Line 39-40: "Milk and its products occupy the first place among the foods present in the daily diet of every human being" This statement is clearly wrong. Large parts of the human population don't consume milk at all, this is very dependent on the country and the culture
Line 49ff: I would recommend to clearly distinguish between contamination of milk as a product and infection of the cow i.e. mastitis
Line 51: bacteria might also originate from the farmer handling the milk
Line 86ff: "On the one hand, the subthreshold concentrations of antibiotics strongly influence the selection of resistant strains, and on the other hand, they contribute to the formation of morphologically changed bacteria" Please provide a reference for this statement
Line 95: "Although microorganisms present in these products do not pose a direct threat to human health" This statement is not true, raw milk can contain many pathogenic bacteria which is why consumption of raw milk (products) is not recommended, especially during pregnancy.
Line 140: Streptococci are actually one of the main causative agents of mastitis so this sentence is misleading
Line 153: "Such bacteria are characterized by the ability to produce lipases, which are responsible for unfavourable changes in the taste and smell of milk." Please provide a reference
Line 171: Mention of fungi is unclear here since you did not search for fungi, did you?
Line 234: You mean the sensitivity of bacteria to beta-lactams? Beta-lactamases will not affect other classes of antibiotics
Line 232: Which new methods do you refer to?
Line 246: What does KO refer to?
Line 246ff: I do not see the relevance of theoretical analyses regarding beta-lactamase presence based only on the 16S rRNA. Many pathogenic bacteria can acquire beta lactamases through plasmids etc and you don't know if this is the case in your samples. Also from line 238ff one would think you looked only at beta-lactamases but in table 1 you also included mecA, please clarify.
Figure 6: see also the comment above. If you only take into account intrinsic Beta-lactamases how can this predict beta-lactam resistance overall?
Line 351: Did you also investigate possible reasons for the observed difference?
Round 2
Reviewer 2 Report
Thanks
Author Response
Dear Reviewer,
Thank you for your additional suggestions on our manuscript.. Please find our answers below. We hope that you will consider our paper ready for publication at this time. Again, thank you very much for your attention.
Conclusions are supported by the results presented earlier in the Results and Discusion section.
The english language of the manuscript was re-verified by a native speaker.
Reviewer 3 Report
I thank the authors for improving the manuscript. There are two points, which I think still need clarification. I added my comments to your responses below.
- Line 86ff: "On the one hand, the subthreshold concentrations of antibiotics strongly influence the selection of resistant strains, and on the other hand, they contribute to the formation of morphologically changed bacteria" Please provide a reference for this statement
Response: We added the reference
Comment: Thank you for adding a reference (15) However I wonder if there is some mistake since the cited paper (15. van Teeseling, M.C.F.; de Pedro, M.A.; Cava, F. Determinants of Bacterial Morphology: From Fundamentals to Possibilities for 440 Antimicrobial Targeting. Front Microbiol 2017, 8, 1264. https://doi.org/10.3389/fmicb.2017.01264) doesn't look at the effect of subthreshold concentrations of antibiotics on selection of resistance
- Line 232: Which new methods do you refer to?
Response: In this sentence, we wanted mention the new genetic methods for the identification of pathogens that can cause mastitis. These methods are still being developed to obtain the best quality and quantity of DNA data and allow for targeting antibiotic treatment. We modified the sentence ”Treatment of mastitis in cows, despite the implementation of prophylactic programs and new genetic methods of diagnosis and treatment, is still mainly based on the administration of antibiotics, despite many negative aspects of their use and not always satisfactory effectiveness”
Comment: I still don't understand what you refer to. Please give an example of a genetic treatment method for mastitis and explain how it is better than administration of antimicrobials.
